# Nutritional Behaviors of Polish Adolescents: Results of the Wise Nutrition—Healthy Generation Project

**DOI:** 10.3390/nu11071592

**Published:** 2019-07-13

**Authors:** Joanna Myszkowska-Ryciak, Anna Harton, Ewa Lange, Wacław Laskowski, Danuta Gajewska

**Affiliations:** 1Department of Dietetics, Faculty of Human Nutrition and Consumer Sciences, Warsaw University of Life Sciences (WULS), 159C Nowoursynowska Str, 02-776 Warsaw, Poland; 2Department of Organization and Consumption Economics, Faculty of Human Nutrition and Consumer Sciences, Warsaw University of Life Sciences (WULS), 159C Nowoursynowska Str, 02-776 Warsaw, Poland

**Keywords:** nutrition, nutritional behavior, diet quality, adolescents

## Abstract

Background: Recognition of the dominant dietary behaviors with respect to gender and specific age groups can be helpful in the development of targeted and effective nutritional education. The purpose of the study was to analyze the prevalence of the selected eating behaviors (favorable: Consuming breakfasts, fruit, vegetables, milk and milk beverages, whole grain bread and fish; adverse: Regular consumption of sweets, sugared soft drinks and fast-foods) among Polish adolescents. Methods: Data on the nutritional behaviors were collected using a questionnaire. Body mass status was assessed based on weight and height measurements. Results: 14,044 students aged 13–19 years old from 207 schools participated in the study. Significant differences were found in the nutritional behaviors depending on age, gender and nutritional status. Favorable nutritional behaviors corresponded with each other, the same relationship was observed for adverse behaviors. The frequency of the majority of healthy eating behaviors decreased with age, whereas the incidence of adverse dietary behaviors increased with age. Underweight adolescents more often consumed sugared soft drinks, sweets and fast food compared to their peers with normal and excessive body mass. Conclusions: A significant proportion of adolescents showed unhealthy nutritional behaviors. Showing changes in the incidence of nutritional behaviors depending on age, gender and body weight status, we provide data that can inform the development of dietary interventions tailored to promote specific food groups among adolescents on different stages of development to improve their diet quality.

## 1. Introduction

The health of children and adolescents is dependent upon food intake that provides sufficient energy and nutrients to promote optimal physical, cognitive and social growth and development [1,2,3]. However, in practice, the implementation of proper nutrition recommendations in these population groups is extremely difficult due to the existing barriers, e.g., availability of healthy food, inadequate nutritional knowledge of caregivers and children and personal food preferences [4,5,6,7]. A great body of the literature indicates the low overall diet quality in children and adolescents, both in terms of the amounts (deficits or excesses) of food/nutrients, and the selection of food groups/food products. One in four Polish 17–18 years old female adolescents did not eat breakfast regularly, and nearly half of them consumed fish only one time per month [8]. Almost 35% of schoolchildren and adolescents aged 9–13 years from rural parts of Poland regularly ate sweets, and 46% failed to consume vegetables and fruit at least once a day [9]. These inadequacies in the assortment and quantities of food products result in an incorrect supply of energy and nutrients. The average European adolescents’ diet is too high in saturated fatty acids and sodium, whereas too low in monounsaturated fatty acids, vitamin D, folate and iodine [10]. In Poland a significant increasing trend in calcium intake in teenagers aged 11–15 years was noted in the last 20 years, but the observed values are still lower than the recommendations [11]. In the US nearly 40% of total energy consumed by two- to 18-year-olds came in the form of empty calories (including 365 kcal from added sugars) [12]. Poor quality of the diet in early life may impair growth and development rate, and also increases the risk of some diet-related diseases (e.g., obesity, type 2 diabetes mellitus, cardiovascular disease and osteoporosis) in the future [3,13].

Although correct nutrition is important throughout the life span, it is possible to distinguish particularly critical periods, i.e., the first 2–3 years [3] and the period of puberty [14,15]. Dramatic physical growth and development during puberty significantly increases requirements for energy, protein, and also others nutrients compared to late childhood. Biological changes related to puberty might significantly affect psychosocial development. Rapid changes in body size, shape and composition in girls might lead to poor body image experience and development of eating disorders [16]. At this age, girls may experience nutritional behaviors leading to weight loss, e.g., alternative diets promoted in the media. Nevertheless, a delay in biological development might lower self-esteem and increase the risk of eating disorders among male teenagers [17]. As young teens are highly influenced by a peer group, then the desire to conform may also affect nutritional behaviors and food intake. Moreover, food choices can be used by adolescents as a way to express their independence from families and parents. At this age, young people may prefer to eat fast-food meals in a peer group instead of meals at home with their families. During middle adolescence (15–17 years) importance of peer groups even raising, and their influence regarding individual food choices peaks. Finally, in the late stage of adolescence (18–21 years) the influence of peer groups decreases, whereas an ability to comprehend how current health behaviors may affect the long-term health status significantly increases [18], which in turn can enhance the effectiveness of nutritional education.

Although nutritional knowledge does not always translate into proper nutritional behavior [19], some data indicate the association between nutritional knowledge and the diet quality among adolescents [20]. Joulaei et al. (2018) observed that an increase in functional nutrition literacy was associated with lower sugar intake and better energy balance among boys and higher dairy intake among girls. Therefore, recognition of the dominant dietary behaviors with respect to gender and specific age groups can be helpful in the development of targeted and effective nutritional education.

In Poland, there are many studies on nutritional behaviors of adolescents [8,9,11], but their limitation is the small number of participants and the lack of representativeness in their selection. The only study involving a large, representative group of Polish adolescents is the health behavior in school-aged children (HBSC) [21], conducted for over 30 years, now in more than 40 countries, including Poland. The HBSC study does not allow us to assess nutritional behaviors of older adolescents, because it covers only the group of 11-, 13- and 15-year-old boys and girls. In Poland there is no research including the wide age range of respondents with all periods of adolescence at the same time and with the same methodology. Therefore, the purpose of the present study was to analyze the frequency of occurrence of the behaviors important in terms of overall diet quality amongst Polish adolescents. The frequency of occurrence of nutritional behaviors was analyzed in the age categories with regard to gender and taking into account the criteria of the weight status.

## 2. Materials and Methods

### 2.1. General Information

The presented study is a part of the research and education program Wise nutrition—healthy generation granted by The Coca-Cola Foundation, and addressed to the secondary and upper secondary school youth, their parents and teachers. The main objective of the program was to educate the secondary and upper secondary school students regarding the importance of healthy nutrition and physical activity in the prevention of the diet-related diseases. The research part of the project included assessing the selected dietary behaviors and parameters related to physical activity of the students and performing anthropometric measurements to assess their nutritional status. Those with diagnosed abnormal body mass were invited to the dietary counseling program (two individual meetings with a dietician). The diagram presenting the overall activities within the project is provided in the Appendix A. Participation in the project was voluntary and totally free of charge for all participants (schools, students and parents). All educational and research activities were carried out in schools participating in the program by trained dieticians. After receiving patronages from the government educational institutions and local authorities, written invitations were sent to all secondary and upper secondary schools in Poland. Nearly 14,000 educational institutions listed in the electronic register of schools of the Minister for National Education were invited to participate. Finally, 2058 schools attended by nearly 450,000 students joined the project in 2013–2015.

This paper focused on the results concerning nutritional behaviors of students (Appendix A). The program was carried out following the standards required by the Helsinki Declaration, and the protocol was approved by the Scientific Committee of the Polish Society of Dietetics. School directors provided written informed consent to participate in the study. Parents were provided with a detailed fact sheet describing the program and had to give written informed consent if they wanted their child to participate. All students over 16 years of age were asked to give their written informed consent to participate in the study.

### 2.2. Study Participants

To examine the selected nutritional behaviors and nutritional status of Polish teenagers, participants were recruited from schools participating in the project. To ensure a representative selection of students, these schools were randomly selected using the stratified sampling method from all of the 2058 enrolled institutions. The sampling was stratified by province and location (large, medium, small city and countryside), as well as the type of school (secondary and upper secondary). Secondary schools (called “gimnazjum” in Poland) are compulsory for all adolescents aged 13–16 years, and are located close to the students’ place of residence. Upper secondary schools (high schools, technical schools and basic vocational schools) include, depending on the type, youth from 16 to 20 years of age. As in the case of secondary schools, students typically live with their families and commute to school. Within the selected schools, as the next step, students were randomly selected from the class registry. Exclusion criteria included: Diagnosed disease that required the use of a special diet, pregnancy or lactation in girls or lack of written consent. All the personal data of participants were fully anonymized. The schools, and consequently, students came from all over Poland, therefore the research was of a nationwide character. In total, 207 schools of the 2058 institutions were enrolled (~10%), and finally 14,044 students participated in the study, including 7553 (53.8%) girls and 6491 (46.2%) boys. The age categories for the studied group were adopted in accordance with the HBSC methodology [21].

### 2.3. Anthropometric Measurements

The assessment of the body weight status of the examined individuals was based on anthropometric measurements (body weight and height) conducted by a trained dietitian. All the measurements were carried out with the equipment provided by The Polish Society of Dietetics: Digital floor scales (TANITA HD-380 BK, Tanita Corporation, Tokyo, Japan) and a steel measuring tape (0–200 cm). All dieticians conducting the measurements were specially trained and followed the same procedures according to Anthropometry Procedures Manual by National Health and Nutrition Examination Survey (NHANES) [22] to minimize bias. The school was obliged to provide a room suitable for the measurements.

Weight of the individuals was measured twice to the nearest 0.1 kg, and the mean value was recorded. Measurements were conducted on individuals dressed in basic clothes (e.g., underwear, trousers/skirt and t-shirt) and without shoes. From the final result 0.5 kg was subtracted (predicted weight of the basic clothes).

For height measurements individuals stood on a flat surface in an upright position with their back against the wall, and the heels together and toes apart (without shoes and socks). They were asked to stand as tall as possible with the head in the Frankfort horizontal plane [22]. The height measurement was conducted twice to the nearest 0.1 cm, and the mean value was recorded.

Based on the body height and weight data, body mass index (BMI) value was calculated. BMI was calculated as body weight in kilograms divided by the square of height in meters. Depending on the age of the subjects different criteria for assessing the body weight status were used. For individuals aged 13–18 years old, calculated BMI value was plotted on gender BMI centile charts for age (with an accuracy of one month) [23]. The percentile value was read from percentile grids and the body mass status was assessed according to the International Obesity Task Force (IOTF) criteria (underweight <5 percentile, normal weight 5–85 percentile, overweight >85 and ≤95 percentile, obese >95 percentile) [24]. For students above the age of 18 years old, the standard World Health Organization (WHO) body mass index criteria were applied: Underweight for BMI <18.5 kg/m^2^, normal body weight for BMI between 18.5 and 24.9 kg/m^2^, overweight between 25 and 29.9 kg/m^2^ and obesity ≥30 kg/m^2^ [25].

### 2.4. Analysis of Nutritional Behaviors

Data on the selected nutritional behaviors were collected prior to the anthropometric measurements and dietary counseling. The paper questionnaire containing questions about the selected nutritional practices, physical activity and self-esteem satisfaction (data not included in this article) was carried out in individuals by a dietitian. This provided the opportunity to clarify possible doubts or ask additional questions. After its completion the questionnaire was collected by a dietician. Due to the large sample group and direct methods of data acquisition, it was decided that the questionnaire has to be short, and must contain questions about the critical determinants of teenagers diet quality. Taking into account the health behavior in school-aged children (HBSC) questionnaire (developed for 11, 13 and 15 year olds) concerning nutritional behaviors [26], and available data on nutritional characteristics of the Polish youth population [21], nine questions were finally formulated with the possibility of answering “yes” or “no”. The first six questions concerned favorable aspects of the nutritional behaviors, while the last three questions referred to the adverse nutritional practices. Healthy nutritional behaviors included: (1) Regular consumption of breakfast before leaving for school, (2) daily consumption of at least one serving of fresh fruit and (3) daily consumption of at least two servings of vegetables (recommended diet quality indicators adapted from HBSC questionnaire [26]. Additionally, taking into account the importance for the overall diet quality and the low consumption in the Polish population [21,27], the three extra questions were added: (4) Daily consumption of milk and/or milk fermented beverages (as the main source of calcium in the diet), (5) daily consumption of whole grains (as the main source of complex carbohydrates and dietary fiber) and (6) consumption of fish at least once week (as the main source of docosahexaenoic acid (DHA), eicosapentaenoic acid (EPA), vitamin D and iodine in the diet). On the other hand, negative dietary determinants (unfavorable nutritional practices increasing the share of free sugars, saturated fat and trans fatty acids in the diet) were considered as: Drinking sugared soft drinks (soda and other carbonated soft drinks) several times during the week, eating sweets more than once a day (adapted from HBSC), and consuming fast food more than twice a week. Prior to the main study, a pilot study (*n* = 50) was conducted to examine whether the questions were understandable to the respondents. The questionnaire was validated: Repeatability was verified by determining the correlation coefficient between the results obtained in the same group (*n* = 50, age 13–19 years old) twice; correlation coefficients for individual questions were on average 0.76 (95% CI = 0.71–0.83) and ranged from 0.18 to 0.96.

### 2.5. Statistical Analysis

Statistical data processing was performed using Statistica version 13.1 (Copyright^©^StatSoft, Inc, 1984–2014, Cracow, Poland). Data were analyzed in the total group, according to age, gender and body weight status. Statistical significances for nominal (categorical) variables were determined using the Pearson’s chi-square test. Additionally, contingency coefficient Cramér’s V was used to indicate the strength of association between categorical variables. Quantitative data was tested for normality of distribution; in the case of its absence the Mann–Whitney test was used for comparisons of independent groups. The correspondence analysis was used to study the relationship between dietary behaviors. The differences were considered significant at *p* ≤ 0.05.

## 3. Results

The total sample group consisted of 14,044 students, including 7553 girls and 6491 boys. The detailed characteristics of the group in terms of age distribution, sex and the body mass index are presented in Table 1. Data on examined dietary behaviors are presented in Table 2. All data are expressed as number values and in percentages.

### 3.1. Characteristics of the Study Group

The characteristics of the study population in terms of age distribution and the body mass index (BMI) in the whole group and separately for girls and boys are presented in Table 1. The predominant group was students aged 17 (followed by 18 and 16 olds in girls, and 16 and 18 olds in boys), while the smallest groups were students aged 13 and 19 between both sex groups. There were significant differences in the average BMI between girls and boys in the total group and in the case of all age categories except the 13 year olds.

### 3.2. Characteristics of Nutritional Behaviors

Figure 1 presents the relationship between the examined nutritional behaviors in the whole group. Based on the correspondence analysis, it is possible to indicate the connections between the analyzed nutritional behaviors. Beneficial nutritional behaviors such as consuming breakfast, fruit, vegetables, whole-grain bread, milk or milk beverages and fish were linked together. In opposite, unfavorable eating behaviors such as skipping breakfast, low consumption of milk products, fruits, vegetables, fish and whole-grain bread were related. Behaviors such as fast food, sweets and sugared soft drinks consumption were linked together and corresponded more to the adverse nutritional behaviors.

The frequencies of examined nutritional behaviors in the total group, and for girls and boys separately are presented in Table 2.

Breakfast was regularly consumed by seven out of 10–13 year olds but only by half of 19 year olds. There was a statistically significant (but small) effect of age in the total group, and separately for girls and boys. Boys were more likely to eat breakfast in comparison to girls, and differences were particularly noticeable in the younger age groups. The frequency of eating at least one serving of fruit per day also decreased with age. A statistically significant (but small) effect in the total group, and separately for girls, and boys was noted. Girls were more likely to include fresh fruits to the daily diet in comparison to boys. There was a significant effect of age on the consumption of vegetables in the total group. Half of the 13-year-olds consumed at least two servings of vegetables a day, but the frequency of consumption decreased to 43% for the group of 19-year-olds. Similarly, the influence of age was observed in the group of girls, and boys. Girls consumed at least two servings of vegetables daily more often than boys in all age groups, except for 18-year-olds. The consumption of milk or milk beverages decreased with age. Significant age effects were observed throughout the total group, and among girls, and boys. In all age groups fewer girls drank milk and fermented milk beverages compared to boys. With age, the proportion of teenagers consuming whole grain bread in everyday diet decreased. However, age effects were not observed for girls, and neither for boys, separately. Considering gender, in all age groups, the greater percentage of girls consumed whole wheat bread in their usual diet. No significant effect of age was observed on fish consumption, neither in the total group, nor in girls, and boys. However, significant gender effects were observed: A greater percentage of boys consumed fish at least one a week in all age groups compared to girls. The effect of age was observed in regards of drinking sugared soft drinks a few times a week, for the whole group, and for both genders. The proportion of adolescents drinking sugared soft drinks increased with age. A higher percentage of boys consumed sugared soft drinks compared to girls in all age categories. Less than half of the students declared consuming sweets more than once a day. No age effects were observed, neither for the whole group, nor for the girls. A small effect of age was found only among boys. On the other hand, a relationship with gender was observed: In all age groups, a higher percentage of girls declared such behavior compared to boys. There was a significant relationship between fast-food consumption and age. The percentage of adolescents consuming fast food more than twice a week increased with age, and an analogous relationship was observed for girls and boys. In all age groups, a higher number of boys declared such nutritional behaviors comparing to girls.

The data on the prevalence of examined nutritional behaviors depending on the nutritional status (underweight, normal body mass, overweight and obesity) are presented in Table 3.

Analyzing the prevalence of selected nutritional behaviors in the whole group of adolescents depending on the body weight status, significant relationships were observed for all eating behaviors except for consuming vegetables. Regular consumption of breakfast was more often declared by adolescents with normal body weight and underweight in total group and both for girls and boys. The percentage of subjects consuming at least one portion of fruit was the smallest in the underweight group, and the largest among the obese adolescents. Consumption of milk and milk beverages was declared by a higher percentage by overweight adolescents, whereas in the smallest percentage by underweight individuals. At the same time, no relationship was observed between this nutritional behavior and the nutritional status separately for girls and boys. The frequency of regular consumption of whole-grain bread increased with the category of body weight in the whole group and in the case of girls. The relationship between fish consumption and the nutritional status was observed only in the whole group. As in the case of bread, the frequency of declared fish consumption increased with the BMI category. In the case of the last three nutritional behaviors: Drinking sweet beverages, eating sweets and fast foods, the incidence of these behaviors decreased with the BMI category, both in the whole group and among girls and boys (with the exception of drinking sweet drinks among boys, where no relation was observed with the body weight status).

## 4. Discussion

Since adolescence is a time of tremendous biological, psychosocial and cognitive changes, nutrition interventions need to be tailored not only to the developmental stage, but also to the nutritional needs of individuals [18]. Based on dietary recommendation, nutritional behaviors crucial for the overall diet quality of children and adolescents might be determined. The “key” determinants of the healthy diet include eating breakfasts, regular consumption of vegetables, fruits, dairy products, whole grain products, fish, as well as avoiding sugared soft drinks, sweets and fast foods (empty calories) [26,28,29]. Literature data indicate the prevalence of selected nutritional behaviors, as well as typical nutritional errors in children and adolescents at different stages of development [8,30,31,32]. Hiza et al. [33] and Bandield et al. [33] reported a poorer diet quality in adolescents compared to younger children. In the US students, a decrease in fruit and vegetable consumption and an increase in fast food intake have been reported from childhood and young adolescence to older adolescence [34]. Nevertheless, Lipsky et al. [29] observed a modest improvement in diet quality between 16.5 and 20.5 years of age reflected, among others, in more frequent breakfasts consumption.

Based on the analysis of correspondence, it can be noticed that regardless of age or sex, beneficial (or adverse) nutritional behaviors cluster together. Thus, individuals who, for example, do not consume breakfast, more often show other adverse nutritional behaviors (a low consumption of fruit, vegetables, fish and whole grain bread). A typical breakfast in Poland includes bread or cereals, dairy and/or meat products as well as vegetables and/or fruit. Thus, omitting breakfast may lead to a reduction in the supply of these products in the overall diet. Our results suggest that if one irregularity is found in a teenager’s diet, it can be assumed that the overall diet quality is low. Interestingly, consuming (or not consuming) sweets, sugared soft drinks and fast food cluster together, but did not correspond to other determinants of the quality of the diet. It may suggest that such products might be consumed together as a meal (e.g., meal typical for fast-food restaurants). It may also suggest the need for educational activities aimed at these products, regardless of the general education about healthy nutrition.

In our study, the frequency of regular breakfast consumption decreased with age, both among boys and girls. In addition, girls significantly less often declared eating breakfast compared to boys. Our observations are consistent with data from the HBSC study [26] where older children and girls were less likely to eat breakfast every weekday. However, more Polish 13- and 15 years olds declared this beneficial nutritional behavior compared to the average among their European peers [26]. Interestingly, we noted a significant relationship between the regularity of consuming breakfast and the body mass status. Regular breakfast consumption was declared by the highest percentage of students with normal body mass (61%) and the lowest with obesity (54%). It could be hypothesized that skipping breakfasts can be a strategy to reduce the weight of adolescents. However, this hypothesis requires additional research. Fayet-Moore et al. [35] observed a lower prevalence of overweight among breakfast consumers compared to skippers (*n* = 4487, 2–16 years). Moreover, individuals who eat breakfast had significantly higher intake of calcium and folate, and significantly lower intake of total fat than breakfast skippers, which indicates the important role of breakfast not only in maintaining a healthy body weight but also in the quality of the diet. Our results indicate a strong need to increase education activities promoting the regular breakfast consumption, especially among older girls and students with abnormal body mass status.

Regular fruit and vegetables consumption is linked to many positive health outcomes [36]. The WHO recommends at least 400 g of fruit and vegetables daily, however studies in 10 European countries indicate that the majority of teenagers fail to meet the recommendations [37]. Only 37% of 13-year-olds and 33% 13-year-olds reported eating fruit at least once a day, whereas vegetables were consumed every day or more than once a day by 35% of the 13-year-olds and 33% of the 13-year-olds, respectively (average from 38 countries and regions) [26]. We observed a decrease in the daily fruit and vegetables consumption with age in the total group, and for both genders; in the total group the percentage of teenagers reporting daily fruit consumption decreased from 67% in 13-years-olds to 49% in 19-years-olds. In the case of vegetables, we did not observe a relationship with body weight status, but the frequency of daily fruit consumption was related to the nutritional state. Regular consumption of fruit was most often declared by obese teenagers, and least frequently by underweight adolescents. The fruit, in contrast to vegetables, have a higher energy value, which, with high consumption, may increase the energy value of the diet. In the case of vegetables and fruit there is still a substantial room for improvement in all subgroups, however education should emphasize differences in the caloric value between fruit and vegetables, especially promoting the latter.

Dairy products, especially milk and milk beverages, contribute to a healthy diet by providing energy, protein, and nutrients such as calcium, magnesium and vitamins B_1_, B_2_ and B_12_ [38]. Regular consumption of at least two servings of dairy products in adolescents resulted in a significant weight loss and a reduction in body fat [39,40]. However, data from HELENA study reported that European adolescents eat less than two-thirds of the recommended amount of milk (and milk products) [37], which reflected in low calcium intake, especially in oldest girls group [10]. We also observed a decrease in the percentage of students declaring daily milk and milk beverages consumption with age. The trend was particularly pronounced among girls: From 56% among 13-year-olds to 43% in 19-year-olds. We also observed a relationship between milk consumption and nutritional status in total group. In this case regular daily milk consumption most often has been declared by individuals with normal body weight. Based on our findings, nutritional education concerning promotion of milk products should be especially targeted at older girls.

As in the case of vegetables and fruits, consumption of whole grain products is associated with a lower risk of many diet-related diseases, e.g., cardiovascular disease and stroke, hypertension, insulin sensitivity, diabetes mellitus type 2, obesity and some types of cancer [41]. Papanikolaou et al. [42] reported a better diet quality and nutrients intake in US children and adolescents consuming grain food products compared to those consuming no grains. In our study less than half of students consumed whole grains bread every day, and the percentage of those decreased with age. Interestingly, the frequency of whole-grain bread consumption was the highest among adolescents with excessive body mass, especially in girls. This may suggest that although consumption of whole grain bread improves the quality of a diet, it may also contribute to increasing the overall caloric value of the diet.

Regular intake of fish, particularly fatty fish, has positive health outcomes, especially in the long term. It reduces the risk of CHD mortality and ischaemic stroke [43]. Fish consumption in adolescents has been associated with better school achievements and performance in cognitive tests [44]. Handeland et al. [45] observed a small beneficial effect of fatty fish consumption on processing speed in tests of attention conducted in 426 students age 14–15 years old. In our study only half of students consumed fish at least once a week, and no age effect has been observed. However, boys declared consumption of fish more often compared with girls. Additionally, the significant relation has been noted between fish consumption and body weight status: The percentage of fish consumers increased with body mass status (45% in underweight and 53% in obese individuals). However, considering the beneficial role of fish and their low intake, nutritional education should be carried out in all subgroup of adolescents, regardless of age, sex or weight status.

Sugared soft drinks, sweets and fast foods are the sources of empty calories that contribute to a substantial share of the total energy intake in children and adolescents [46]. Intake of soft (sweetened) drinks among adolescents is higher than in other age groups (nearly 20% of 13- and 15-years olds reported their regular daily consumption), and it is associated with a greater risk of weight gain, obesity and chronic diseases and directly affects dental health by providing excessive amounts of sugars [26]. In our study, consumption of sugared soft drinks increased with the age category in total group, and both for boys and girls, but in the same time was the lowest in the case of obese individuals compared to other weight groups (except for boys). Sweetened beverages provide a high-energy amount in liquid form that contributes to increasing the simple-carbohydrate content of the diet and influencing the other nutrients’ intake [12,47]. Interestingly, similar relationships were also observed in the case of fast food consumption. While sweets consumption was significantly higher in girls and underweight students, but no effect of age has been noted. The HBSC data also highlighted gender differences in daily sweets intake (27% of 13-years old girls compared to 23% of boys in the same age). Taking into account the prevalence of these adverse behaviors, nutritional education should be directed at all adolescents, but with particular focus on older age groups.

### Strengths and Limitations

The strength of the study is the sample size. To our knowledge there is no research on such a scale covering all age categories over a large geographic area. With such a large sample, the advantage is also the way of obtaining data. All questionnaires were filled in by a trained dietician who could explain the respondents’ doubts on an ongoing basis. Moreover, all anthropometric data were obtained through measurements conducted also by a dietician, which ensured obtaining reliable results and minimize the bias.

Respondents for our study were recruited from schools participating in the project, which can be a certain limitation. However, the number of schools allowed a random selection of the sample taking into account different types of institutions and their geographic location. The small number of questions with the very limited possibilities of answers in the questionnaire may also be a certain limitation. However, the questions have been developed on the basis of large, international studies on the nutritional behaviors of school-aged children [21,26], and include the most important healthy and unhealthy behaviors concerning nutrition. Additionally, the questionnaire was validated before the main study.

## 5. Conclusions

By analyzing the differences in nutritional behaviors between age and gender groups, we provide data that can inform the development of dietary interventions tailored to answer the needs of adolescents at different stage of development and to improve the quality of their diet. We observed significant changes in the frequencies of analyzed eating behaviors depending on gender as well as on age. Furthermore, we have shown that the incidence of undesirable eating behavior is higher among underweight adolescents compared to their peers with an excessive body mass. Information on the most frequent nutritional errors on every stage of adolescents might be used to determine the type of educational messages given when counseling this challenging group, e.g., education activities regarding regular breakfast consumption should be intensified in older age groups, as the percentage of young people who eat breakfast decreases with age. On the other hand, education on the adverse effects of consumption of sweets, sugared soft drinks and fast food should be directed not only to adolescents with excessive body weight, but mainly to those underweight, as the consumption of these products is more frequent in this group. Moreover, regardless of age and sex, both favorable and adverse nutritional behaviors corresponded with each other. The present findings can be used both for the development of educational programs and for educational activities carried out by teachers at the school level.

## Figures and Tables

**Figure 1 nutrients-11-01592-f001:**
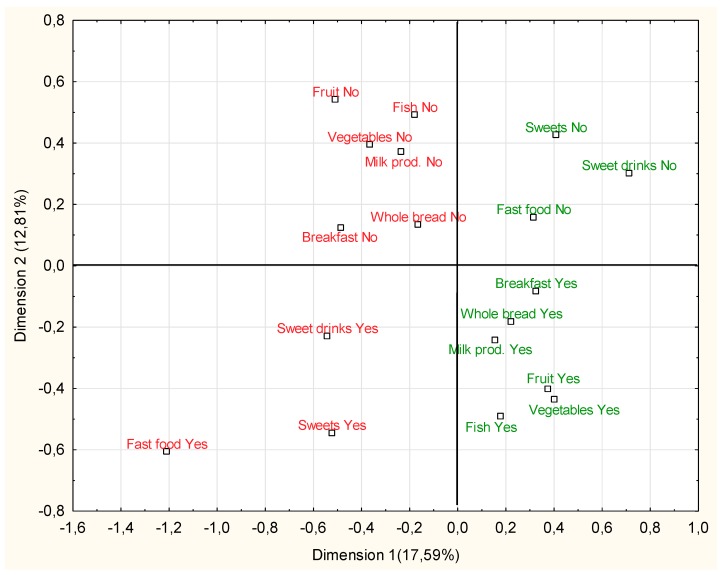
The results of the analysis of correspondence of all examined variables (nutritional behaviors) in the total group. Behaviors beneficial to the overall diet quality are marked in green; unfavorable behaviors are marked in red.

**Table 1 nutrients-11-01592-t001:** Age and body mass index (BMI) distribution of the individuals: In the total group and divided by gender.

	Total Group			Girls			Boys			*p*-Value
Age (years)	*N*	%	BMI mean ± SD	*N*	%	BMI mean ± SD	*N*	%	BMI mean ± SD	
13	1270	9.0	20.1 ± 3.66	670	8.9	19.9 ± 3.55	600	9.2	20.3 ± 3.78	0.0854
14	1876	13.4	20.5 *±* 3.48	1004	13.3	20.3 ± 3.41	872	13.4	20.6 ± 3.54	0.0414 ^1^
15	2011	14.3	20.8 *±* 3.37	1040	13.8	20.6 ± 3.17	971	15.0	21.0 ± 3.56	0.0485 ^1^
16	2386	17.0	21.3 ± 3.31	1257	16.6	21.1 ± 3.08	1129	17.4	21.6 ± 3.54	0.0024 ^1^
17	2781	19.8	21.8 *±* 3.48	1538	20.4	21.3 ± 3.30	1243	19.2	22.3 ± 3.61	0.0000 ^1^
18	2551	18.2	21.9 *±* 3.41	1436	19.0	21.3 ± 3.16	1115	17.2	22.7 ± 3.56	0.0000 ^1^
19	1169	8.3	22.2 *±* 3.61	608	8.1	21.7 ± 3.68	561	8.6	22.6 ± 3.48	0.0000 ^1^
All age groups	14044	100.0	21.3 ± 3.51	7553	53.8	20.9 ± 3.33	6491	46.2	21.7 ± 3.68	0.0000 ^1^

^1^ Significant differences in average BMI between girls and boys for age categories and for the total (all age groups), the Mann–Whitney test.

**Table 2 nutrients-11-01592-t002:** Nutritional behaviors of the individuals: In the total group and divided by gender.

Age Category (years)	Total Group		Girls		Boys		*p*-Value (V Cramer) Girls vs. Boys
	*N*	%	*N*	%	*N*	%	
	Having breakfast every day before leaving for school						
All age groups	8400	59.81	4178	55.32	4222	65.05	
13	897	70.63	431	64.33	466	77.67	0.0000 ^4^ (0.10)
14	1222	65.14	595	59.26	627	71.90	
15	1267	63.00	587	56.44	680	70.03	
16	1405	58.89	682	54.26	723	64.04	
17	1620	58.25	836	54.36	784	63.12	
18	1391	54.53	751	52.30	640	57.40	
19	598	51.15	296	48.68	302	53.83	
*p*-value	0.0000 ^1^		0.0001 ^2^		0.0000 ^3^		
V Cramer	0.10		0.08		0.14		
	Consuming fresh fruit every day (at least one serving)						
All age groups	8071	57.47	4506	59.66	3565	54.93	
13	857	67.48	468	69.85	389	64.83	0.0000 ^4^ (0.05)
14	1242	66.20	686	68.33	556	63.76	
15	1267	63.00	673	64.71	594	61.17	
16	1322	55.41	746	59.35	576	51.02	
17	1490	53.58	865	56.24	625	50.32	
18	1322	51.82	751	52.30	571	51.21	
19	571	48.85	317	52.14	254	45.28	
*p*-value	0.0000 ^1^		0.0000 ^2^		0.0000 ^3^		
V Cramer	0.13		0.13		0.13		
	Consuming vegetables every day (at least two servings)						
All age groups	6678	47.55	3658	48.43	3019	46.52	
13	662	52.13	372	55.52	290	48.33	0.0241 ^4^ (0.02)
14	1010	53.84	561	55.88	449	51.49	
15	996	49.53	533	51.25	463	47.68	
16	1130	47.36	614	48.85	516	45.70	
17	1245	44.77	698	45.38	547	43.96	
18	1135	44.51	616	42.90	519	46.56	
19	500	42.77	264	43.42	236	42.07	
*p*-value	0.0000 ^1^		0.0000 ^2^		0.0061 ^3^		
V Cramer	0.0		0.07		0.05		
	Drinking milk or milk beverages (yoghurt/kefir/butter milk, etc.) every day						
All age groups	8478	60.38	4172	55.24	4306	66.36	
13	860	67.72	438	65.37	422	70.33	0.0000 ^4^ (0.11)
14	1279	68.18	643	64.04	636	72.94	
15	1305	64.89	610	58.65	695	71.58	
16	1392	58.34	674	53.62	718	63.60	
17	1583	56.92	780	50.72	803	64.65	
18	1435	56.27	729	50.80	706	63.32	
19	624	53.42	298	49.01	326	58.21	
*p*-value	0.0000 ^1^		0.0000 ^2^		0.0000 ^3^		
V Cramer	0.10		0.11		0.09		
	Consuming whole-grained bread every day						
All age groups	5962	42.46	3302	43.72	2659	40.97	
13	556	43.78	297	44.33	259	43.17	0.0010 ^4^ (0.03)
14	809	43.12	447	44.52	362	41.51	
15	871	43.33	467	44.95	404	41.61	
16	1065	44.64	578	45.98	487	43.14	
17	1198	43.08	679	44.15	519	41.71	
18	1020	39.98	598	41.64	422	37.85	
19	443	37.90	236	38.82	207	36.90	
*p*-value	0.0007 ^1^		*p* = 0.0566		*p* = 0.0548		
V Cramer	0.04						
	Consuming fish at least once a week						
All age groups	7023	50.02	3412	45.19	3610	55.63	
13	662	52.13	341	50.90	321	53.50	0.0000 ^4^ (0.1)
14	965	51.44	463	46.12	502	57.57	
15	1021	50.80	469	45.10	552	56.91	
16	1188	49.81	564	44.90	624	55.27	
17	1330	47.84	673	43.79	657	52.82	
18	1260	49.39	626	43.59	634	56.86	
19	597	51.07	276	45.39	321	57.22	
*p*-value	0.1057		0.0640		0.2130		
V Cramer							
	Drinking sugared soft drinks few times a week						
All age groups	7977	56.80	3764	49.83	4212	64.90	
13	650	51.18	305	45.52	345	57.50	0.0000 ^4^ (0.15)
14	1069	56.98	512	51.00	557	63.88	
15	1202	59.77	540	51.92	662	68.18	
16	1374	57.59	633	50.36	741	65.63	
17	1546	55.59	719	46.75	827	66.51	
18	1441	56.49	720	50.14	721	64.66	
19	695	59.45	335	55.10	360	64.17	
*p*-value	0.0000 ^1^		0.0026 ^2^		0.0019 ^3^		
V Cramer	0.05		0.05		0.06		
	Consuming sweets more than once a day						
All age groups	6156	43.84	3428	45.39	2727	42.02	
13	548	43.15	311	46.42	237	39.50	0.0000 ^4^ (0.03)
14	840	44.78	465	46.31	375	43.00	
15	926	46.05	486	46.73	440	45.31	
16	1065	44.65	573	45.62	492	43.58	
17	1216	43.73	690	44.86	526	42.27	
18	1068	41.87	628	43.73	440	39.46	
19	493	42.17	275	45.23	218	38.86	
*p*-value	*p* = 0.0912		*p* = 0.7863		0.0480 ^3^		
V Cramer					0.04		
	Consuming fast food (e.g., chips, burgers etc.) more than two times a week						
All age groups	2906	20.70	1360	18.01	1545	23.81	
13	206	16.22	99	14.78	107	17.83	0.0000 ^4^ (0.07)
14	314	16.74	145	14.54	168	19.27	
15	385	19.14	170	16.35	215	22.14	
16	513	21.50	226	17.98	287	25.42	
17	622	22.37	299	19.44	323	25.93	
18	590	23.14	295	20.54	295	26.48	
19	276	23.63	125	20.56	151	26.96	
*p*-value	0.0000 ^1^		0.0002 ^2^		0.0000 ^3^		
V Cramer	0.06		0.06		0.07		

^1^ Significant differences in the examined nutritional behaviors between age categories in the total group; ^2^ significant differences in the examined nutritional behaviors between age categories in girls; ^3^ significant differences in the examined nutritional behaviors between age categories in boys; ^4^ significant differences in the examined nutritional behaviors between girls and boys, the chi^2^ Pearson test.

**Table 3 nutrients-11-01592-t003:** Nutritional behaviors of the individuals depending on body weight status: In the total group and divided by gender.

BMI Category	Total Group		Girls		Boys	
	*N*	%	*N*	%	*N*	%
	Having breakfast every day before leaving for school					
Underweight	422	59.44	256	55.65	166	66.40
Normal	6567	60.99	3302	56.28	3265	66.62
Overweight	908	55.64	407	52.38	501	58.60
Obesity	502	53.98	212	47.53	290	59.92
*p* value	0.0000 ^1^		0.0012 ^2^		0.0000 ^3^	
V Cramer	0.05		0.04		0.06	
	Consuming fresh fruit every day (at least one serving)					
Underweight	360	50.70	233	50.65	127	50.80
Normal	6108	56.72	3467	59.09	2641	53.89
Overweight	994	60.91	499	64.22	495	57.89
Obesity	606	65.16	305	68.39	301	62.19
*p* value	0.0000 ^1^		0.0000 ^2^		0.0006 ^3^	
V Cramer	0.06		0.07		0.05	
	Consuming vegetables every day (at least two servings)					
Underweight	334	47.04	214	46.52	120	48.00
Normal	5086	47.23	2828	48.20	2258	46.07
Overweight	799	48.99	384	49.42	415	48.59
Obesity	457	49.14	230	51.57	227	46.90
*p* value	0.4228		0.4139		0.5479	
V Cramer						
	Drinking milk or milk beverages (yoghurt / kefir / butter milk, etc.) every day					
Underweight	392	55.21	233	50.65	159	63.60
Normal	6538	60.72	3266	55.68	3272	66.76
Overweight	1000	61.27	429	55.21	571	66.78
Obesity	545	58.67	241	54.04	304	62.94
*p* value	0.0174 ^1^		0.2010		0.2830	
V Cramer	0.03					
	Consuming whole-grained bread every day					
Underweight	285	40.14	177	38.48	108	43.20
Normal	4522	42.00	2547	43.42	1975	40.30
Overweight	713	43.69	352	45.30	361	42.22
Obesity	440	47.31	224	50.22	216	44.63
*p* value	0.0059 ^1^		0.0031 ^2^		0.1984	
V Cramer	0.03		0.04			
	Consuming fish at least once a week					
Underweight	323	45.49	193	41.96	120	52.00
Normal	5356	49.75	2632	44.88	2176	55.59
Overweight	849	52.02	370	47.62	376	56.02
Obesity	493	53.01	216	48.43	207	57.23
*p* value	0.0071 ^1^		0.1157		0.5947	
V Cramer	0.03					
	Drinking sugared soft drinks few times a week					
Underweight	435	61.27	260	56.52	175	70.00
Normal	6140	57.02	2943	50.16	3197	65.23
Overweight	901	55.21	361	46.46	540	63.16
Obesity	499	53.66	199	44.62	300	61.98
*p* value	0.0096 ^1^		0.0007 ^2^		0.1098	
V Cramer	0.03		0.05			
	Consuming sweets more than once a day					
Underweight	403	56.76	272	59.13	131	52.40
Normal	4900	45.51	2751	46.89	2149	43.85
Overweight	572	35.05	278	35.78	294	34.39
Obesity	280	30.14	127	28.54	153	31.61
*p* value	0.0000 ^1^		0.0000 ^2^		0.0000 ^3^	
V Cramer	0.11		0.13		0.09	
	Consuming fast food (e.g., chips, burgers etc.) more than two times a week					
Underweight	187	26.34	115	25.00	72	28.80
Normal	2311	21.46	1090	18.58	1221	24.92
Overweight	274	16.80	105	13.51	169	19.79
Obesity	134	14.41	50	11.21	84	17.36
*p* value	0.0000 ^1^		0.0000 ^2^		0.0000 ^3^	
V Cramer	0.06		0.07		0.06	

^1^ Significant differences in the examined nutritional behaviors in the total group; ^2^ significant differences in the examined nutritional behaviors between girls; ^3^ significant differences in the examined nutritional behaviors between boys; the chi^2^ Pearson test.

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
