# Peer review of "Nutritional Behaviors of Polish Adolescents: Results of the Wise Nutrition—Healthy Generation Project"

_nutrients, 2019, doi:10.3390/nu11071592_

Round 1

Reviewer 1 Report

Thank you for the opportunity to review this manuscript which describes dietary quality among Polish adolescents, considering differences by age, sex, and weight status. The topic provides an addition to the literature as no other studies have described dietary quality of Polish adolescents across this large age range and within such a large sample. I have noted below some specific areas for clarification that I think will strengthen the paper. In addition, the paper would benefit from a thorough editing for word choice and grammar.

Abstract

·        Lines 24-25: both favorable and adverse nutritional behaviors corresponded with each other—this wording makes it seem like favorable and adverse behaviors were related to each other, but what I think is meant is that behaviors within each of these categories were related? Please reword/clarify

Introduction:

·        Lines 55-69: I appreciate the discussion of developmental changes during adolescence, but this paragraph doesn’t feel connected to the topic of the paper. Can the authors add a sentence or two more indicating the importance of these developmental changes to dietary intake specifically?

·        Line 79: the authors note they will be considering “key behaviors” in terms of overall diet quality. Please note what these are, and why they are important to diet quality.

Method

·        Line 90: The authors note that individuals with diagnosed abnormal body mass were invited to dietary counseling. It’s unclear to me from the figure whether this counseling happened before or after the measures used in this study were collected (or if these individuals were a part of the sample in this study). Please clarify. If measures were collected after the counseling, how might this have affected outcomes?

·        Lines 163-164: Why were only 3 questions used for adverse nutritional practices whereas 6 were used for favorable?

·        Nutritional behaviors: Either in the methods, or in the introduction, please address why these specific behaviors (favorable and adverse) are most important for overall dietary quality.

·        Lines 174-175: I’m not sure what you mean by “check the correctness of the formulated questions”. Please clarify.

Discussion

·        Lines 286-288: issues with references

Author Response

Response to Reviewer 1 Comments

R1: Thank you for the opportunity to review this manuscript which describes dietary quality among Polish adolescents, considering differences by age, sex, and weight status. The topic provides an addition to the literature as no other studies have described dietary quality of Polish adolescents across this large age range and within such a large sample. I have noted below some specific areas for clarification that I think will strengthen the paper. In addition, the paper would benefit from a thorough editing for word choice and grammar.

Abstract

R1:  Lines 24-25: both favorable and adverse nutritional behaviors corresponded with each other—this wording makes it seem like favorable and adverse behaviors were related to each other, but what I think is meant is that behaviors within each of these categories were related? Please reword/clarify

Authors' response: Thank you for the remark. Yes, These behaviors corresponded within the category (favourable corresponded with favourable and adverse corresponded with adverse). It has been clarified in the manuscript (marked in blue). It reads: “Favourable nutritional behaviors corresponded with each other, the same relationship was observed for adverse behaviors.”

Introduction:

R1: Lines 55-69: I appreciate the discussion of developmental changes during adolescence, but this paragraph doesn’t feel connected to the topic of the paper. Can the authors add a sentence or two more indicating the importance of these developmental changes to dietary intake specifically?

Authors' response: According to the suggestion, this is included in the text (marked in blue).

R1: Line 79: the authors note they will be considering “key behaviors” in terms of overall diet quality. Please note what these are, and why they are important to diet quality.

Authors' response: We partially agree that the term “key behaviors” might be unclear in this point of manuscript. All behaviors (which were the subject of questions) are described in the Materials and methods section. Following the comment we modified the sentence and it now reads: “Therefore, the purpose of the present study was to analyze the frequency of occurrence of the behaviors important in terms of overall diet quality amongst Polish adolescents.”

Method

R1: Line 90: The authors note that individuals with diagnosed abnormal body mass were invited to dietary counseling. It’s unclear to me from the figure whether this counseling happened before or after the measures used in this study were collected (or if these individuals were a part of the sample in this study). Please clarify. If measures were collected after the counseling, how might this have affected outcomes?

Authors' response: The body weight and height measurements and the questionnaire on nutritional behaviours were conducted prior to the dietary counselling. Based on anthropometric measurements, individuals with abnormal body mass were identified and possibility of dietary counselling were offered to them (it is indicated in the graph by directions of arrows).  

It is stated in the manuscript (2.3. Anthropometric Measurements, line 133): “The assessment of the body weight status of the examined individuals was based on anthropometric measurements (body weight and height) conducted by a trained dietitian”; and in the next paragraph (section 2.4. Analysis of Nutritional Behaviors): “Data on the selected nutritional behaviors were collected prior the anthropometric measurements.”

R1: Lines 163-164: Why were only 3 questions used for adverse nutritional practices whereas 6 were used for favorable?

Authors' response: It was not the intention of the authors to obtain the same number of questions in relation to beneficial and unfavourable behaviors. However, for practical reasons (a very large group and a time limit to conduct a survey), we tried to limit the total number of questions to a minimum. When formulating questions, the model was based on the HBSC questionnaire (research conducted in over 40 countries, for over 30 years), which is beneficial, among others for the possibility of data comparison.  The HBSC questionnaire contains 5 questions, including 3 for favorable (breakfast, fruit, vegetables) and 2 for adverse (sweets, sweetened soft drinks). The questions we added resulted from the recommendations of proper nutrition addressed to children and adolescents, often presented in the form of a pyramid, i.e. consumption of wholemeal bread, milk and fermented milk beverages and fish, and also literature review on nutritional behaviors of Polish students. We decided to add a question about the consumption of fast-food foods, which can be asses as an adverse eating behaviors (common in this age group). Detailed information about selecting questions are in the Materials and methods section, analysis of nutritional behaviors.

R1: Nutritional behaviors: Either in the methods, or in the introduction, please address why these specific behaviors (favorable and adverse) are most important for overall dietary quality.

Authors' response: Following the remark, this information has been added (marked in blue).

R1: Lines 174-175: I’m not sure what you mean by “check the correctness of the formulated questions”. Please clarify.

Authors' response: During the pilot study, it was verified whether the questions are understandable to the respondents, e.g. whether determining the whole-wheat bread is known in this age group. It has been clarified in the manuscript (marked in blue).

Discussion

R1:  Lines 286-288: issues with references

Authors' response: The missing references have been added (marked in blue). “Literature data indicate the prevalence of selected nutritional behaviors, as well as typical nutritional errors in children and adolescents at different stages of development [8,30,31,32]. Hiza et al. [33] and Bandield et al. [33] reported a poorer diet quality in adolescents compared to younger children.”

Thank you very much for your time and all the valuable comments.

Reviewer 2 Report

In the article, "Nutritional behaviors of Polish adolescents: results of the Wise nutrition – healthy generation project," the authors present the results of a large survey of Polish adolescents. The sample size is impressive, and the article is generally well-written. There are just a few comments that would improve the manuscript, detailed below:

The introduction could be reorganized to make it more clear what the article is about. Similarly, it is unclear how the current study is different than the presented Polish data.

The aims could be written more clearly, making it clear that the data will be examined by sex and weight status.

Is it standard practice to remove a standard weight to account for clothing?

It is unclear how age was calculated - was anyone 13-13.999 considered 13? Rounded?

The discussion would benefit from more explanation of the findings. For example, why do you think those of higher weight statuses are more likely to skip breakfast? In an attempt to lose weight, etc.

Author Response

Response to Reviewer 2 Comments

R2: In the article, "Nutritional behaviors of Polish adolescents: results of the Wise nutrition – healthy generation project," the authors present the results of a large survey of Polish adolescents. The sample size is impressive, and the article is generally well-written. There are just a few comments that would improve the manuscript, detailed below:

R2: The introduction could be reorganized to make it more clear what the article is about. Similarly, it is unclear how the current study is different than the presented Polish data.

Authors' response: We partially agree with the comment. The introduction section has been reorganized, according to the comments of other reviewer and suggested information has been added. It reads: “In Poland, there are many studies on nutritional behaviors of adolescents [8,9,11], but their limitation is the small number of participants and the lack of representativeness in their selection. The only study involving a large, representative group of Polish adolescents is the Health  Behaviour  in  School-aged  Children (HBSC) [21], conducted for over 30 years, now in more than 40 countries, including Poland. The HBSC study does not allow to assess nutritional behaviors of older adolescents, because it covers only the group of 11, 13 and 15 year olds. In Poland there is no research including the wide age range of respondents with all periods of adolescence at the same time and with the same methodology. Therefore, the purpose of the present study was to analyze the frequency of occurrence of the key behaviors in terms of overall diet quality amongst Polish adolescents. The frequency of occurrence of nutritional behaviors was analyzed in the age categories with regard to gender and taking into account the criteria of the weight status.”

R2: The aims could be written more clearly, making it clear that the data will be examined by sex and weight status.

Authors' response: We agree with the comment and missing information has been added (all changes introduced into the manuscript have been marked in green). It now reads: “Therefore, the purpose of the present study was to analyze the frequency of occurrence of the key behaviors in terms of overall diet quality amongst Polish adolescents. The frequency of occurrence of nutritional behaviors was analyzed in the age categories with regard to gender and taking into account the criteria of the weight status.

R2: Is it standard practice to remove a standard weight to account for clothing?

Authors' response: Yes, it is a standard procedure in Poland, recommended by specialists, e.g. by The Polish Society of Dietetics (in Polish: Zespół ds. leczenia otyłości u osób dorosłych Polskiego Towarzystwa Dietetyki: Gajewska D, Myszkowska-Ryciak J, Lange E, Gudej S, Pałkowska-Goździk E, Bronkowska M, Piekło B, Łuszczki E, Kret M; Białek-Dratwa A, Pachocka L, Sobczak-Czynsz A. Standardy leczenia dietetycznego otyłości prostej u osób dorosłych. Stanowisko Polskiego Towarzystwa Dietetyki 2015. Dietetyka 2015 vol. 8, Wyd. Spec.). Is also practised and described in other countries, e.g.  De  Groot  C.P.G.M.,  Sette  S.,  Zajkas  G.,  et  al.:  Nutritional  status:  anthropometry.  Eur.  J.  Clin.Nutr. 1991, 45, (suppl. 3), 31.

R2: It is unclear how age was calculated - was anyone 13-13.999 considered 13? Rounded?

Authors' response: Age categories were adopted according to the procedure described in the international HBSC studies (cited in the manuscript). This information has been added to the manuscript. It reads (marked in green): “The age categories for the studied group were adopted in accordance with the HBSC methodology [21].”

In the case the nutritional status assessment, the percentile value on the BMI chart for each subject was determined with an accuracy of 1 month (separately for girls and boys). This is described in the materials and methods section, anthropometric measurements, lines 144-145 original manuscript version: “For individuals aged 13 - 18 years old, calculated BMI value was plotted on gender BMI centile charts for age (with an accuracy of one month) [22].”

R2: The discussion would benefit from more explanation of the findings. For example, why do you think those of higher weight statuses are more likely to skip breakfast? In an attempt to lose weight, etc.

Authors' response: We partially agree with the comment and some more information in discussion section has been added. However, the main purpose of the manuscript was to present data on the nutritional behaviors of adolescents in a large, representative group. We are currently preparing a manuscript discussing factors affecting examined behaviors and their interrelationships. Therefore, in the present article such relationships were not discussed in detail.  

Thank you very much for your time and all the valuable comments.

Reviewer 3 Report

The topic is interesting, but as it stands, the article needs more information on how sugar-sweetened beverages were defined, how the questionnaire was validated, which criteria were used to classify adolescents of different ages based on their weight, and if the nutritional data among students with unhealthy body weights was collected before they received individual dietary counseling in the form of a dietary intervention. This information is crucial to determine the validity of the data reported in the manuscript. There were also many mistakes throughout the manuscript, I have highlighted the major ones.

Major Comments

1) Please always use the same term to allude to sugar-sweetened beverages. As it is, the authors alternate between the terms “sweet drinks”, “sweet beverages” and “sugary drinks”. Also, the authors need to define what beverages they included in this category as the definition can vary from one country to another. Was it soft drinks, sports drinks, energy drinks, fruit drinks with added sugar, flavored milk with added sugar, slush, coffees and teas with added sugar, etc.? Did they include diet soft drinks? Did they include 100% pure fruit juices?

2) Line 143 on page 3 and line 254 on page 9: I am not sure if the authors mean nutritional status or weight status as weight is used to classify adolescents. Please clarify this information and please use the same term to allude to this as the authors alternate between “nutritional status” and “nutrition status” (line 276 on page 11), and also “body mass status” and “body mass index”.

3) Lines 173-175 on page 4: The authors need to add more information on how the questionnaire used to collect data on adolescents’ nutritional behaviors was validated. For example, was the pilot study conducted among a similar population in terms of age, gender, ethnicity, etc.? Did the authors validate their questionnaire against another validated questionnaire, such as a food-frequency questionnaire, a food record and/or a 24-hour dietary recall (i.e., criterion validity)? Did the authors calculate Cronbach alphas to verify the internal consistency of their questionnaire? Was the temporal stability (a type of reliability) of their questionnaire evaluated in a test-retest study? This information is crucial to determine the psychometric qualities of the questionnaire used to collect data and to determine the validity of the study.

4) Table 3 on pages 9-10: Please clarify how adolescents were classified as underweight, normal weight, overweight and with obesity as different criteria were used based on age. As it stands, it is not clear if BMI > 25 or 30 kg/m2 or the percentile or a mix of both criteria were used. This could be added as a footnote at the bottom of the table.

5) Line 303 on page 11: This sentence is contradictory. The authors mentioned that sweets, sugar-sweetened beverages and fast-food might be consumed together as separate meals. How can these food be consumed together if it is at separate meals?

6) Lines 315-316 on page 11: The authors do not report in the present paper adolescents’ nutrients intake. I therefore suggest them to not refer to this data in their discussion or at least, to mention that this data is not shown.

7) Line 392 on page 13: I do not understand this sentence. Please reformulate.

8) Lines 398-399 on page 13: Before the authors can mention that having a validated questionnaire is a strength of their study, they need to add more information on how this validation was performed. As it stands, it looks like they only tested the comprehension of their questionnaire, which is a very low level of validation. If so, this would be considered a weakness of their study.

9) Based on the supplementary material, it is not clear if the data on dietary habits was collected before the students with unhealthy body weights received individual dietary counseling in the form of a dietary intervention. This information is crucial as the intervention could have changed the dietary habits of students with unhealthy body weight and bias the results reported in the article.

Minor Comments

1) Line 46 on page 1: There is a mistake, please replace “ones” by “once” (i.e., once a day).

2) Line 54 on page 2: Please give examples of diet-related diseases. Do the authors mean overweight/obesity, type 2 diabetes and/or hypertension?

3) Line 62 on page 2: There is a mistake, please replace “males” by “male” (i.e., male teenagers).

4) Line 6 on page 2: There is a mistake, please replace “individuals” by “individual” (i.e., individual food choices).

5) Lines 70-73 on page 2: Please define what is meant by nutritional literacy.

6) Line 79 on page 2: Please state which key behaviors will be examined in the study.

7) Line 102 on page 3: There is a mistake, please replace “schools” by “school” (i.e., school directors).

8) Lines 167 and 172 on page 4: I am not sure if the authors mean adapted rather than adopted from the HBSC.

9) Line 179 on page 4: I am not sure if the authors mean quantitative rather than qualitative variables. I am not sure how Pearson’s chi-square test can be used to analyze qualitative data.

10) Line 218 on page 6: There is an omission, please add a hyphen between “10” and “13” (i.e., 10-13 year olds).

11) Line 289 on page 11: The authors used “USA” while they previously used “US”. Please always use the same abbreviation to refer to the United States.

12) Line 303 on page 11: There is a mistake, please replace “consume” by “consumed” (i.e., might be consumed together).

13) Line 403 on page 13: There is a mistake, please remove the extra “their” between “quality” and “of” (i.e., the quality of their diet).

14) Supplementary material: There is a mistake, please replace “form” by “from” in the square on recruitment of the schools and the students (i.e., consent from parents or legal guardians of adolescents). Also, why is there a dotted line around the square on questionnaire on nutritional habits? Please add a footnote at the bottom of the figure to explain this.

Author Response

Response to Reviewer 3 Comments

R3: The topic is interesting, but as it stands, the article needs more information on how sugar-sweetened beverages were defined, how the questionnaire was validated, which criteria were used to classify adolescents of different ages based on their weight, and if the nutritional data among students with unhealthy body weights was collected before they received individual dietary counseling in the form of a dietary intervention. This information is crucial to determine the validity of the data reported in the manuscript. There were also many mistakes throughout the manuscript, I have highlighted the major ones.

Authors' response: We appreciate all comments and suggestions. Changes made by us in the manuscript are highlighted in brown.

Major Comments

R3 1) Please always use the same term to allude to sugar-sweetened beverages. As it is, the authors alternate between the terms “sweet drinks”, “sweet beverages” and “sugary drinks”. Also, the authors need to define what beverages they included in this category as the definition can vary from one country to another. Was it soft drinks, sports drinks, energy drinks, fruit drinks with added sugar, flavored milk with added sugar, slush, coffees and teas with added sugar, etc.? Did they include diet soft drinks? Did they include 100% pure fruit juices?

Authors' response: We agree that the definition of beverages may vary from country to country, as well as in the studies of individual authors. Quoting literature, we tried to keep the terminology used by the authors of the publication, what we think is the most correct procedure.

However, we agree fully with the comment about the need to define the definition in our study. This information has been added in the material and methodology section and the term: "sugared soft drinks" has been used in the manuscript. Milk drinks, tea and coffee, diet drinks, fruit juices were not counted as sugared soft drink.

R3 2) Line 143 on page 3 and line 254 on page 9: I am not sure if the authors mean nutritional status or weight status as weight is used to classify adolescents. Please clarify this information and please use the same term to allude to this as the authors alternate between “nutritional status” and “nutrition status” (line 276 on page 11), and also “body mass status” and “body mass index”.

Authors' response: These terms are often used to describe similar states, and even interchangeably. However, as suggested, we have arranged the terminology leaving the term "body weight status".

R3 3) Lines 173-175 on page 4: The authors need to add more information on how the questionnaire used to collect data on adolescents’ nutritional behaviors was validated. For example, was the pilot study conducted among a similar population in terms of age, gender, ethnicity, etc.? Did the authors validate their questionnaire against another validated questionnaire, such as a food-frequency questionnaire, a food record and/or a 24-hour dietary recall (i.e., criterion validity)? Did the authors calculate Cronbach alphas to verify the internal consistency of their questionnaire? Was the temporal stability (a type of reliability) of their questionnaire evaluated in a test-retest study? This information is crucial to determine the psychometric qualities of the questionnaire used to collect data and to determine the validity of the study.

Authors' response: An assessment of the accuracy of the measurement was used (the ability of the questionnaire to measure a given feature in such a way that the value obtained by means of the measurement was consistent with the actual value [in Polish: Jędrychowski W.: Metoda zbierania wywiadów lekarskich i budowa kwestionariuszy zdrowotnych. PZWL, Warszawa, 1982]. In assessing the accuracy of the questionnaire, repeatability was used. Repeatability of the questionnaire was verified by determining the correlation coefficient between the results obtained for the same individuals in the first and second interview (n = 50, determining intra-individual variability). For almost all questions, a high correlation was found between the results obtained in both interviews. The correlation coefficients for individual questions were on average 0.76 (95% CI: 95% CI = 0.71-0.83) and ranged from 0.18 to 0.96).

R3 4) Table 3 on pages 9-10: Please clarify how adolescents were classified as underweight, normal weight, overweight and with obesity as different criteria were used based on age. As it stands, it is not clear if BMI > 25 or 30 kg/m2 or the percentile or a mix of both criteria were used. This could be added as a footnote at the bottom of the table.

Authors' response: We cannot agree with the comment as the methodology of classification (underweight, normal weight, overweight and obese) has been fully described in section 2.3. Anthropometric Measurements:

"Based on the body height and weight data, body mass index (BMI) value was calculated. BMI was calculated as body weight in kilograms divided by the square of height in meters. Depending on the age of the subjects different criteria for assessing the nutritional status were used. For individuals aged 13 - 18 years old, calculated BMI value was plotted on gender BMI centile charts for age (with an accuracy of one month) [23]. The percentile value was read from percentile grids and the body mass status was assessed according to the International Obesity Task Force (IOTF) criteria (underweight < 5 percentile, normal weight 5 - 85 percentile, overweight > 85 and ≤ 95 percentile, obese > 95 percentile) [24]. For students above the age of 18 years old, the standard World Health Organization (WHO) body mass index criteria were applied: underweight for BMI < 18.5 kg / m2, normal body weight for BMI between 18.5 and 24.9 kg / m2, overweight between 25 and 29.9 kg / m2, and obesity ≥ 30 kg / m2 [25]."

The BMI formula is the same for all participants. But the way in which BMI values are interpreted depends on age - this is accepted and described in the quoted literature.

R3 5) Line 303 on page 11: This sentence is contradictory. The authors mentioned that sweets, sugar-sweetened beverages and fast-food might be consumed together as separate meals. How can these food be consumed together if it is at separate meals?

Authors' response: We agree with the comment and now it reads: " It may suggest that such products might be consumed together as a meal (e.g. meal typical for fast-food restaurants).

R3 6) Lines 315-316 on page 11: The authors do not report in the present paper adolescents’ nutrients intake. I therefore suggest them to not refer to this data in their discussion or at least, to mention that this data is not shown.

Authors' response: We cannot agree with the comment. These information is cited to illustrate / emphasize the important role of breakfast in the diet quality. Therefore, we think that they should stay in the text.

R3 7) Line 392 on page 13: I do not understand this sentence. Please reformulate.

Authors' response: It now reads: " Respondents for our study were recruited from schools participating in the project, which can be a certain limitation".

R3 8) Lines 398-399 on page 13: Before the authors can mention that having a validated questionnaire is a strength of their study, they need to add more information on how this validation was performed. As it stands, it looks like they only tested the comprehension of their questionnaire, which is a very low level of validation. If so, this would be considered a weakness of their study.

Authors' response: Not only the understanding of the questions by the respondents was checked but also the repeatability of the results after two repetitions on the same sample of respondents. We can add that this questionnaire is used by us to assess basic nutritional behaviors. We added more information about the procedure of validation in the manuscript.

R3 9) Based on the supplementary material, it is not clear if the data on dietary habits was collected before the students with unhealthy body weights received individual dietary counseling in the form of a dietary intervention. This information is crucial as the intervention could have changed the dietary habits of students with unhealthy body weight and bias the results reported in the article.

Authors' response: The data on dietary habits was collected before the students with unhealthy body weights received individual dietary counseling. In our opinion it is clearly marked with the direction of the arrows on the diagram in supplemetary materials. Additionally, it is stated in the manuscript (2.4. Analysis of Nutritional Behaviors): "Data on the selected nutritional behaviors were collected prior the anthropometric measurements and dietary counselling ".

Minor Comments

R3 1) Line 46 on page 1: There is a mistake, please replace “ones” by “once” (i.e., once a day).

Authors' response: This has been corrected.

R3 2) Line 54 on page 2: Please give examples of diet-related diseases. Do the authors mean overweight/obesity, type 2 diabetes and/or hypertension?

Authors' response: The examples of diet-related diseases have been added.

 R3 3) Line 62 on page 2: There is a mistake, please replace “males” by “male” (i.e., male teenagers).

Authors' response: This has been corrected.

R3 4) Line 6 on page 2: There is a mistake, please replace “individuals” by “individual” (i.e., individual food choices).

Authors' response: This has been corrected.

R3 5) Lines 70-73 on page 2: Please define what is meant by nutritional literacy.

Authors' response: Based on the quoted source, we have changed the term "nutritional literacy" to: "nutritional knowledge", which is better recognized.

R3 6) Line 79 on page 2: Please state which key behaviors will be examined in the study.

Authors' response: Due to the suggestion of the another reviewer, this sentence has been modified and now reads: "Therefore, the purpose of the present study was to analyze the frequency of occurrence of the behaviors important in terms of overall diet quality amongst Polish adolescents. The frequency of occurrence of nutritional behaviors was analyzed in the age categories with regard to gender and taking into account the criteria of the weight status. "

R3 7) Line 102 on page 3: There is a mistake, please replace “schools” by “school” (i.e., school directors).

Authors' response: This has been corrected.

R3 8) Lines 167 and 172 on page 4: I am not sure if the authors mean adapted rather than adopted from the HBSC.

Authors' response: Examples mentioned above relate to different situations. In the first case (line 167): questions from the HBSC questionnaire are developed for 11, 13 and 15 years (for these age groups were validated). To clarify it has been changed in the manuscript. We agree that in line 172 the term "adapted" is more appropriate.

R3 9) Line 179 on page 4: I am not sure if the authors mean quantitative rather than qualitative variables. I am not sure how Pearson’s chi-square test can be used to analyze qualitative data.

Authors' response: We are sure we used the proper test for our date. According statistics manual, qualitative variables (also known as nominal / categorical variables), are variables with no natural sense of ordering. They are therefore measured on a nominal scale. For instance, the answer "yes" or "no" for question about regular consumption of breakfast is a qualitative variable. Qualitative variables can be coded to appear numeric but their numbers are meaningless, as in yes=1, no=2. The Chi Square statistic is commonly used for testing relationships between categorical variables. [Statistica, StatSoft, 13.1, https://statistics.laerd.com/spss-tutorials/chi-square-test-for-association-using-spss-statistics.php].

R3 10) Line 218 on page 6: There is an omission, please add a hyphen between “10” and “13” (i.e., 10-13 year olds).

 Authors' response: This has been corrected

R3 11) Line 289 on page 11: The authors used “USA” while they previously used “US”. Please always use the same abbreviation to refer to the United States.

Authors' response: This has been corrected.

R3 12) Line 303 on page 11: There is a mistake, please replace “consume” by “consumed” (i.e., might be consumed together).

 Authors' response: This has been corrected.

R3 13) Line 403 on page 13: There is a mistake, please remove the extra “their” between “quality” and “of” (i.e., the quality of their diet).

Authors' response: This has been corrected.

R3 14) Supplementary material: There is a mistake, please replace “form” by “from” in the square on recruitment of the schools and the students (i.e., consent from parents or legal guardians of adolescents). Also, why is there a dotted line around the square on questionnaire on nutritional habits? Please add a footnote at the bottom of the figure to explain this.

Authors' response: This has been corrected. Dotted line around the square on questionnaire on nutritional habits has been removed.

Thank you very much for your time and all the valuable comments.

Round 2

Reviewer 3 Report

The authors took into account most of the reviewers' comments and the manuscript has improved as a result. The article is now suitable for publication in Nutrients after a grammar check.